# Knowledge and Awareness of Algerian Students about Cervical Cancer, HPV and HPV Vaccines: A Cross-Sectional Study

**DOI:** 10.3390/vaccines10091420

**Published:** 2022-08-29

**Authors:** Djihad Bencherit, Rania Kidar, Saadia Otmani, Malik Sallam, Kamel Samara, Hiba Jawdat Barqawi, Mohamed Lounis

**Affiliations:** 1Department of Biology, Faculty of Natural and Life Sciences, University of Ziane Achour, Djelfa 17000, Algeria; 2Department of Pathology, Microbiology and Forensic Medicine, School of Medicine, The University of Jordan, Amman 11942, Jordan; 3Department of Clinical Laboratories and Forensic Medicine, Jordan University Hospital, Amman 11942, Jordan; 4Department of Translational Medicine, Faculty of Medicine, Lund University, 22184 Malmö, Sweden; 5College of Medicine, University of Sharjah, Sharjah 27272, United Arab Emirates; 6Department of Clinical Sciences, College of Medicine, University of Sharjah, Sharjah 27272, United Arab Emirates; 7Department of Agro-Veterinary Sciences, Faculty of Natural and Life Sciences, University of Ziane Achour, Djelfa 17000, Algeria

**Keywords:** cervical cancer, HPV infections, HPV vaccines, Algeria

## Abstract

Cervical cancer is one of the most recurrent and dangerous female cancers in Algeria and worldwide. It is mainly caused by human papillomaviruses (HPV), which can induce other cancers as well. Although it can be fatal, cervical cancer is one of the most preventable and manageable cancers. While HPV vaccination is the key weapon to win the battle against this pathology, it is still not approved in Algeria. Therefore, we set up a cross-sectional survey to assess the knowledge and awareness of students from Algerian universities concerning cervical cancer and HPV and to understand their attitudes towards HPV vaccination. The results reveal that, out of 715 students, the majority of students were aware of cervical cancer (84.6%). However, only 46.2% of students had heard about HPV. Furthermore, willingness to get the HPV vaccine was estimated at 26.7% among students with prior knowledge of HPV, and 21.5% of these students claimed to be willing to pay to get the HPV vaccine if available. Nevertheless, HPV vaccine hesitancy was estimated at 37.5% among the students aware of HPV. The main causes of HPV vaccine reluctance were: complacency (30.6%), refusal of vaccination in general (20.2%) and belief in the rarity of HPV-induced infections in Algeria (19.4%). Moreover, the embrace of vaccine conspiracy beliefs among students were significantly related with their HPV vaccine rejection. Overall, these findings encourage the establishment of a social education policy concerning the fight against HPV-induced cancers, particularly that of the cervix, and the implementation of a national HPV vaccination program targeting young people.

## 1. Introduction

Being the 4th most common cancer striking women’s health worldwide, cervical cancer affects half a million women each year, with half of those affected dying of the disease [1,2]. According to the World Health Organization (WHO), 90% of cervical cancer deaths happen in developing countries. In fact, in 2016, 85% of all cervical cancer cases were recorded in developing countries [2]. Even with the low incidence in North Africa, cervical cancer holds the second rank on the podium of cancers threatening the health of women in Algeria and Morocco and the 3rd rank in Tunisia [3].

Cervical cancer is mainly caused by viruses from the human papillomaviruses family, HPV. It is a family of double-stranded circular DNA viruses (8 Kb) with a tropism for cutaneous and mucosal epithelia [4,5]. These viruses are mainly transmitted sexually, with HPV being the most recurrent sexually transmitted disease [2]; however, the possibility of cutaneous contamination (intimate skin to skin contact) cannot be excluded [5,6].

The HPV family has more than 200 members classified into two groups according to their pathogenicity. Non-oncogenic HPVs, also referred to as low-risk HPV, are associated with recurrent respiratory infections and anogenital warts. Conversely, oncogenic HPVs, also known as high-risk HPV, are responsible for chronic HPV infections with a high potential for anogenital cancers (penile, anal, cervical, vaginal and vulvar) and head and neck squamous cell carcinomas (oral cavity, oropharyngeal and laryngeal and other pharynx cancers) in both sexes [1,4,5,6,7]. Among more than 14 oncogenic HPVs, the HPV16 and 18 genotypes are the most redundant among HPV-induced cancers and are involved in more than 70% of the cervical precancerous lesions, cervical cancers and other HPV-associated cancers [5]. HPV strains 31, 33, 45, 52 and 58 are also oncogenic, being implicated in 10% to 20% of cervical cancer [8].

Hence, vaccination presents an opportunity to greatly diminish HPV-induced cancers threatening both sexes, especially cervical cancer in females. There are three HPV vaccines, namely Cervarix 2v (GlaxoSmithKline Biologicals, Rixensart, Belgium), Gardasil 4v (Merck&Co, Kenilworth, NJ, USA) and Gardasil 9v (Merck&Co, Kenilworth, NJ, USA) [1,5,8]. These are recombinant HPV vaccinal proteins of each virus genotype expressed on the surface of viruses like particles, which makes the vaccine effective against several viral genotypes at the same time (National Cancer Institute, 2021; Szymonowicz and Chen, 2020). Cervarix, for example, protects against genotypes 16 and 18. Gardasil 4v confers protection against HPV 6 and 11, in addition to HPV16 and HPV 18. Compared to the quadrivalent, the nonavalent version of Gardasil (Gardasil 9v) confers additional protection against HPV genotypes 31, 33, 45, 52 and 58 [5,8]. The latter is the only vaccine used in the USA since 2016, while Gardasil 4v and Cervarix continue to be used in other countries [8].

Recommended by the WHO, HPV vaccines are implemented in the vaccination schedule of 105 countries. According to the recommendations of the WHO, HPV vaccination is mainly recommended for young adults of both sexes aged 26 or lower, given their great vulnerability to HPV infections [8]. Except the United Arab Emirates, no Arab country has undertaken a vaccination program targeting HPV, despite a prevalence of 16% in the general population and 80% among cervical cancer samples [9]. In Algeria, HPV vaccination is not part of the vaccination program and HPV vaccines are not approved [10]. 

Hence, this survey targets students of Algerian universities to estimate their knowledge about cervical cancer and HPV viruses, as well as to understand their attitudes towards vaccination, both generally and related to HPV.

## 2. Materials and Methods

### 2.1. Study Design

In this study, a self-administered questionnaire (Appendix A) was developed based on previous studies about HPV and cervical cancer knowledge and attitudes towards HPV vaccination, especially among college students [11,12,13]. It was developed in Arabic and French in Google Forms (Google LLC, Menlo Park, CA, USA, 2021) and was disseminated online using social media platforms between 15 March and 15 May 2022 in accordance with the Strengthening of the Reporting of Observational Studies in Epidemiology (STROBE) guidelines for cross-sectional studies [14].

The target population was all Algerian students living and studying in Algeria regardless of major. The eligibility criteria included: (i) students aged over 18 years old and (ii) the ability to communicate in Arabic or French. Inversely, non-resident students and those who could not communicate in Arabic or French were excluded. Participants were invited to give their consent before enrolling, knowing that participation was voluntary and without any incentives. Collected data was kept and analyzed confidentially.

The minimum sample size was calculated using the Check-Market online sample-size calculator [15]. The number was 664 according to the following assumptions: a total of 1,800,000 students in the Algerian universities for the academic year of 2021/2022, a 5% margin of error and a 95% confidence interval.

### 2.2. Survey Items

The survey consisted of five sections: first, a short introduction describing the objectives of the study and an item to establish participant’s agreement to participate in this study. The second section collected the demographic and the educational qualifications of the participants (age, sex, marital status, university, school/faculty, living area). In the third section, participants were asked if they had heard about cervical cancer and HPV, as well about their sources of information. Those who heard about cervical cancer were asked to about its origin and its diagnosis; as for those who had heard about HPV, they were asked an eight-item binary knowledge scale. Each correct response was scored as one, while the incorrect responses were scored as zero, yielding an HPV knowledge score varying from zero to eight. The fourth section evaluated the attitude of the participants towards HPV vaccine and delineated causes of vaccine reluctance. Finally, for the last section, a seven-item previously validated vaccine conspiracy beliefs scale (VCBS) was used with a 7-point Likert scale to evaluate for vaccine beliefs [16].

### 2.3. Statistical Analysis 

Data was processed in python using Matplotlib-v3.4.2, pandas-v1.2.4 and statsmodels-v0.12.2. Given the optional nature of most questions, there were a proportion of missing data for each question. However, given that demographics had no missing values, this did not affect the analysis discussed later. First, categorical variables were explored using percentages, frequencies and simple graphs for variables like prior knowledge of HPV and cervical cancer, while mean and standard deviation (SD) were used to assess the HPV knowledge and VCBS score. Data was unified across the various languages, and any non-standard responses were grouped into others. All demographic variables were used as predictors of having heard of HPV and cervical cancer. The variable set had no continuous variables. Bivariate analyses were conducted to identify significant predictors using Chi-squared tests. The cut-off for significance was a P-value less than 0.05. All significant predictors were fed into a bivariate logistic regression model, which was evaluated using a likelihood-ratio test.

## 3. Results

### 3.1. Demographic and Scholar Features of the Respondents

In the current study, a total of 720 responses were obtained, reduced to 715 after the removal of 5 responses due to incompleteness. Demographic characteristics of the sample showed that most of the respondent students were single (91.9%, n = 657), female (75.9%, n = 542), living in urban areas (89.1%, n = 637) and aged between 20 and 29 years (74.7%, n = 534). For the scholar features, results showed that students of natural and life sciences (33.6%, n = 240) and humanities (28.4%, n = 203) faculties were the most represented in the sample. Medical studies students only constituted 7.6% of the total sample. The results also showed that the levels of Bachelor’s (46.9%, n = 335) and Master’s (46.9%, n = 335) degrees were equally represented, followed by post-graduate students (6.3%, n = 45). Table 1 presents the distributions of all of the demographic variables.

### 3.2. Cervical Cancer Knowledge

Out of the 715 students, 84.6% (n = 605) reported having previously heard of cervical cancer. The highest level of prior knowledge was found among medical (94.4%, n = 51) and natural and life sciences students (90.4%, n = 217), married persons (96.6%, n = 56), those aged over 30 years (93.3%, n = 70), females (88.4%, n = 479) and Master’s (89.3%, n = 299) and post-graduate degree holders (88.9%, n = 40). Yet, only about 13% (n = 92) of students cited knowledge regarding the cause of HPV, with less than half mentioning sexual relations as a cause. The results of the logistic regression confirmed that females (OR = 2.759, 95% CI: 1.728–4.406), students aged over 30 years (OR = 3.530; 95% CI: 1.053–11.834), those with Master’s degrees (OR = 1.853; 95% CI: 1.119–3.071) and those from the medical and natural and life sciences faculties have higher odds of having heard of cervical cancer compared to their counterparts (Table 2). 

When asking about early screening, 26.6% (n = 161) of students declared that they were aware of the availability of early screening tests for cervical cancer, while 13.38% (n = 81) of students declared the inverse. A total of 40.4% (n = 65) of students reported that the cervical smear was the main early screening tests for cervical cancer. Only 1.25% (n = 9) of female students declared that they did the cervical smear. As for the minimal uptake of screening, most reasons were related to a lack of an awareness campaign about this disease (20.5%, n = 102), negligence (18.9%, n = 94), fear (5.2%, n = 26), shame (5.2%, n = 26), being not married (5%, n = 25) and complacency (2%, n = 10).

Finally, for sources of information, the respondent students who had a prior knowledge about cervical cancer were mainly informed through Internet/social media platforms (43.1%, n = 261), media (radio, television/newspapers) (20.8%, n = 126) and family/friends communication (16.5%, n = 100) (Figure 1).

### 3.3. HPV Prior Knowledge

Regarding prior knowledge about HPV, results showed that only 46.2% (n = 330) of students have heard about this virus. The highest level of prior HPV knowledge was reported among medical (77.8%, n = 42) and natural and life sciences students (58.8%, n = 141). Furthermore, females (50.7%, n = 275), students aged more than 30 years (54.7%, n = 41), those who were post graduation (51.1%, n = 23), those who are married (50%, n = 29) and those living in an urban area (48%, n = 306) have a higher prior knowledge about HPV than their counterparts. The binary logistic regression showed that females (OR = 1.70; OR: 1.161–2.489) studying in the natural and life science faculty and living in urban areas (OR = 2.04; OR: 1.197–3.469) are more likely to have prior knowledge about HPV (Table 3).

One hundred and thirty respondents (39.4%, n = 130) declared that they know the diseases that HPV can cause. The most cited diseases were cervical cancer (60%), cancers in general (18.5%), vaginal cancer (13.1%), penile cancer (12.3%), genital warts (12.3%) and oropharyngeal cancer (10.8%). Results showed that Internet/social media (45.5%), university courses (20.6%), media (13.3%) and family/friends communication (13.3%) were the main source of information about HPV among students who have previously heard about this virus (Figure 1).



$e^{\beta_{i}}$


$\mathrm{Intercept}\;(\beta_{0}$



### 3.4. HPV Level of Knowledge

Among the students who have prior knowledge of HPV, the medium knowledge of HPV was estimated at 64.7%. The highest level of knowledge was obtained for the items: HPV can cause cervical cancer (93.1%) and HPV is sexually transmitted (82.3%), and the lowest levels were obtained for the items: individuals can be infected by HPV for years without knowing (19.6%) and are you aware of the availability of HPV vaccines? (39.5%) (Figure 2).

We farther classified students with prior knowledge about HPV according to the 9-item HPV knowledge scale, and the results showed that the mean score is 6.33 (SD = 1.75). The highest score was recorded among medical students (7.46, SD = 1.78), post-graduate students (7.87, SD = 1.80) and females (7.6, SD = 1.75).

### 3.5. Attitude towards HPV vaccination

Among the total student respondents with a prior knowledge of HPV, only 5.8% (n = 19) have received the HPV vaccine, 26.7% (n = 88) declared themselves willing to receive the HPV vaccine, 21.5 % (n = 71) are willing to pay for the vaccine and 37.6% (n = 124) declared themselves unwilling to obtain vaccines and 33.3% (n = 110) did not answer the question.

The most cited causes for being against HPV vaccination were “I don’t consider myself at risk of HPV infection” (30.6%, n = 38), “I’m against vaccines in general” (20.2%, n = 25) and “I don’t consider HPV as a common infection in Algeria” (19.4%, n = 24) (Figure 3).

### 3.6. Attitude of Participants towards General Vaccination

The results of the respondents about vaccines in general showed that, generally, a high number of them still believe conspiracy theories about vaccines. Figure 4 shows the repartition of the student’s answers for each item. The results of the VCBS for the participants who had heard of HPV was 24.44 (SD = 9.86). The VCBS score was higher among the students who rejected the HPV vaccine (25.77, SD = 9.84) compared to those who were willing to get it (mean: 22.69, SD = 9.74) (*p* = 0.0407).

## 4. Discussion

Cervical cancer is the 4th most common cancer in women worldwide and the 2^nd^ most common cancer threatening women’s lives in developing countries. The global fight against this disease begins with awareness campaigns, screening programs and, above all, vaccination uptake. Thereby, this work aimed to assess the level of knowledge of Algerian university students about cervical cancer, HPV and HPV vaccination.

Apparent was an extremely high level of awareness (84.6%) among students regarding cervical cancer, similar to the level of awareness reported by a study carried out among Indian students [12]. An almost equivalent percentage was recorded (about 81%) among Chinese women in the general population of China [17]. Locally, a survey conducted by our follow Algerian doctors reported a similar level of awareness, about 74% among Algerian women over 25 years old. Compared to our survey, the slight variation in the percentage of awareness among the Algerian population could be mainly explained by the fact that the study of Belhadj et al. was intended for women over 25 years old with different grade levels, of whom only 26.98% were university graduates [11]. An almost equivalent awareness rate (70%) was found among women aged 31 to 45 in Saudi Arabia [18]. However, only 43% of women in the western region of this country had ever heard of this pathology [19]. 

Almost all of the surveyed female students (88.4%) were aware of cervical cancer. This female predominance in cervical cancer knowledge was also reported among students in India (82.4%) [12]. This finding is basically linked to the fact that cervical cancer is a gender pathology and is the 2nd most common cancer affecting women in developing countries [1,2]. In addition to being a girl, we also noticed that the level of cervical cancer awareness was high among married students (96.6 %) and among students in medical fields (94.4%), in natural and life sciences (90.4%), in Master’s degree programs (89.3%) and post-graduation (88.9%), as well as among students over the age of 30 (93.3%). The results of the logistic regression showed that the parameters of gender, age over 30 years, being a student in a Master’s degree program or in the natural and life sciences are significantly associated with a high level of knowledge of cervical cancer.

Although demonstrating a high level of awareness about cervical cancer (84.6%), only 26.6% of the surveyed female students were aware of the availability of a cervical cancer screening test, and about 14% were aware of the PAP SMEAR test. Among women in the general population, a higher level of awareness about the cervical cancer screening test was previously described in Algeria (41.57%) and Saudi Arabia (between 43.5% and 52.5%) [11,18,19]. This almost doubly high level of knowledge and awareness of the cervical cancer screening test could arguably be attributed to the difference in the age range of the target population in the different surveys compared to the current study. Despite the burden of cervical cancer in Algeria, 13.39% of girls surveyed admitted their ignorance of the screening test. This denial has often been justified here, as in the survey by Belhadj et al., by a lack of knowledge and recklessness [11]. Other arguments were also claimed, such as shyness, being unmarried and complacency. Apart from that, the absence of a national cervical cancer screening program and the cost of the screening test would undoubtedly be among the main obstacles to being screened against this pathology in Algeria. The scarcity of means of awareness and prevention against cervical cancer was also reflected in the students’ answers concerning the source of information about cervical cancer, given that almost half (43.1%) cited the Internet and social media platforms, while the rest mentioned the media (20.8%) and immediate environment communication (16.5%).

Moreover, despite the high level of awareness recorded among students, gaps in knowledge on cervical cancer were found, since not even half of the students mentioned sexual relations as a risk factor in infection by cervical cancer and since 87% were unaware that HPV contamination is the main etiology in cervical cancer development. A better level of knowledge was described among students in Beijing, with around 53% making the link between HPV and cervical cancer [20]. Furthermore, more than half of young Hungarian students knew that HPV infection could lead to cervical cancer [21]. However, if45.61% of female students in India established the link between HPV viruses and cervical cancer, only 27.62% of their male colleagues did so [12]. Similarly, a poor level of knowledge has been described among female students in Nigeria, as only 11% of them identified HPV as the etiology for developing cervical cancer [22].

Intriguingly, 46.2% of students are aware of HPV without necessarily linking it to cervical cancer. Herein, we also reported that the parameters being a girl, studying natural and life sciences and living in an urban area are associated with having prior knowledge of HPV virus. Limited awareness about HPV was also described across the population of women in other Arabic countries such as Bahrain (31.3%), Egypt (33.2%) and Saudi Arabia (34.5%) [23]. Other studies have described even poorer knowledge of HPV among the general population in Bahrain (13.5%) and Saudi Arabia (7.2%) [18,24]. Conversely, the work of Sallam et al. in Jordan and that of Ortashi et al. in the United Arab Emirates reported a higher level of awareness of HPV, reaching 72% and 97%, respectively, among female students engaged in medical fields training, most likely because of the medical background of their university courses [13,25]. On the other hand, works carried out on the general population in China showed that about a third of the surveyed population had already heard of HPV [17,26]. Our results are in line with the level of HPV awareness described among female students in India (45.61%) and China (59.1%) [12,27]. Other surveys performed in China revealed a better level of awareness about HPV (70.8% and 76.7%) [20,28]. However, only 17.7% of education and social sciences female students in Nigeria had heard about HPV [22]. 

Although vaccination is the best, even, the only glimmer of hope to thwart HPV-induced cancers, especially cervical cancer, a poor knowledge of HPV vaccine (39.5%) was found among the surveyed students, reflecting the state of awareness of the students about HPV infections. This level of HPV vaccine awareness is similar to that described among female university students in health schools in Jordan (48.7%), among Indian college students (44%) and among university students in west China (56.3%) [12,13,27]. However, the majority (72.6%) and almost all (97%) Beijing college students and students of the school of nurses in the United Arab Emirates, respectively, were aware of the availability of HPV vaccination [20,23]. A poor level of HPV vaccine awareness was found, however, among female students in Nigeria [22]. Our results are comparable to the level of knowledge about HPV vaccination described among women in Saudi Arabia (32.3%) or in China (20%) [17,23,24]. This state of awareness about HPV vaccination turned out to be higher among women in Lebanon (63.5%) [23,29]. However, only 3.9% of women surveyed by Dhaher et al. in Saudi Arabia had heard of HPV vaccination [19].

One of the most striking results of our study is that almost a third of students with prior knowledge about HPV are willing to get the vaccine, even if HPV vaccines are not approved in Algeria. Furthermore, 21.5% of the students agree to pay for the vaccine if available. Despite the fact that this level of acceptance to the HPV vaccine is encouraging, it seems to be almost three-fold lower than that recorded among students from the school of nurses in the United Arab Emirates (74%) and than that of health schools in Jordan (75%) and female students of education and social sciences in Nigeria (about 58%) [13,22,23]. In western China, 65.7% of university students were willing to be vaccinated against HPV [27]. 

Moreover, though 26.7% of students with HPV prior knowledge are willing to get the HPV vaccine if locally approved, about 38% of them refuse to be vaccinated, citing several reasons. For example, about 31% don’t consider themselves at risk to be HPV-exposed, and almost 20% are against vaccination in general; about 19% believe that HPV infection is not very widespread in Algeria. Furthermore, students having HPV vaccine conspiracy beliefs are more likely to report vaccine rejection (*p* = 0.04). Such a correlation had also been reported by Sallam et al. [13]. Furthermore,, it would be important to emphasize, here, that our results are not surprising, considering the absence of an HPV vaccine policy and the absence of awareness campaigns about the fight against HPV-induced cancers. Additional reasons for the reluctance of HPV vaccination have been mentioned around the world. Students at Western China University, for example, reported the cost of the vaccine (51%), worry about side effects (about 46%) and lack of a sexual life (43.4%) as HPV vaccination obstacles [27]. However, complacency was the major factor of vaccine rejection reported by female university students in Jordanian health schools [13]. Across the general population in mainland China, safety concerns, the lack of knowledge and the cost of vaccination were the main arguments given by students when asked about their hesitation towards HPV vaccination [26].

## 5. Conclusions

To sum up, these findings suggest a limited to poor level of awareness about HPV and the HPV vaccine among Algerian university students, even if their level of cervical cancer awareness was satisfying. Taken together, these results highlight the fundamental need to set up a national awareness plan about cervical cancer, HPV and HPV vaccination using modern communication skills, particularly social media platforms. Furthermore, the integration of HPV vaccination in the vaccination schedule and the approval of this vaccination in Algeria are urgent requesst to help decrease the risk of HPV-related diseases in the current post-secondary-school-aged population.

## Figures and Tables

**Figure 1 vaccines-10-01420-f001:**
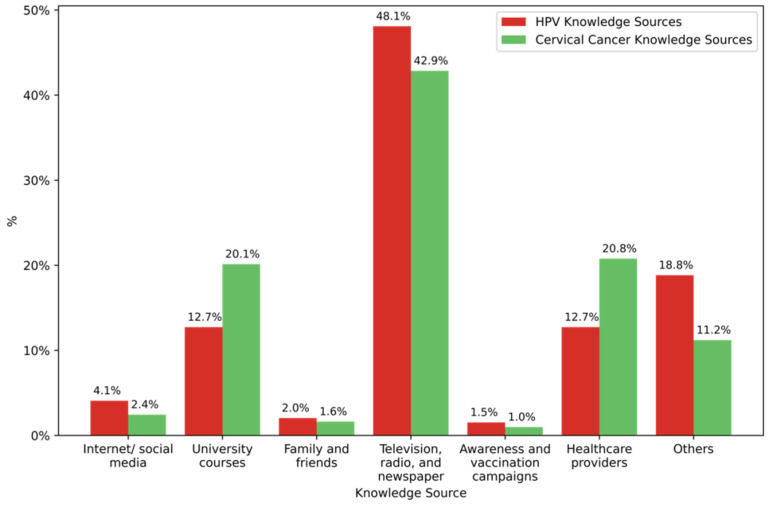
The different knowledge sources used by the participants to learn about cervical cancer and HPV.

**Figure 2 vaccines-10-01420-f002:**
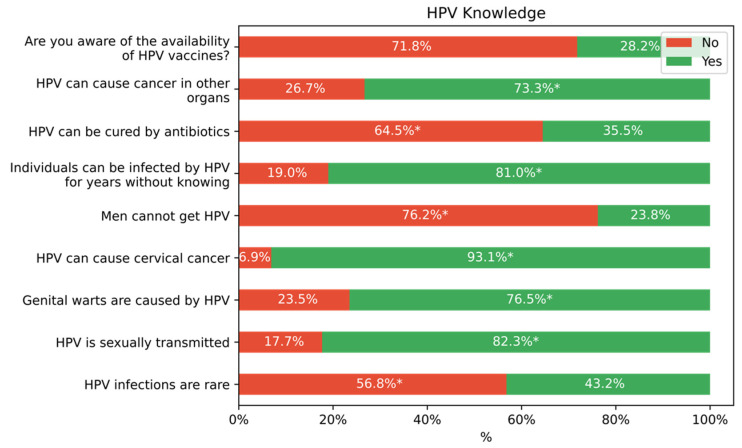
The distribution of responses to the questions evaluating HPV knowledge. * indicates correct responses.

**Figure 3 vaccines-10-01420-f003:**
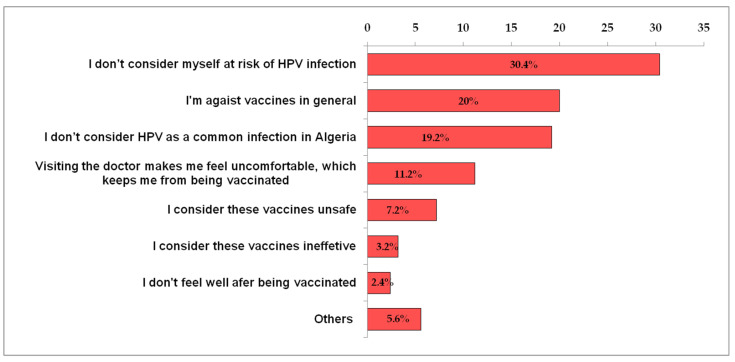
Cause of the rejection of HPV vaccines among the Algerian students.

**Figure 4 vaccines-10-01420-f004:**
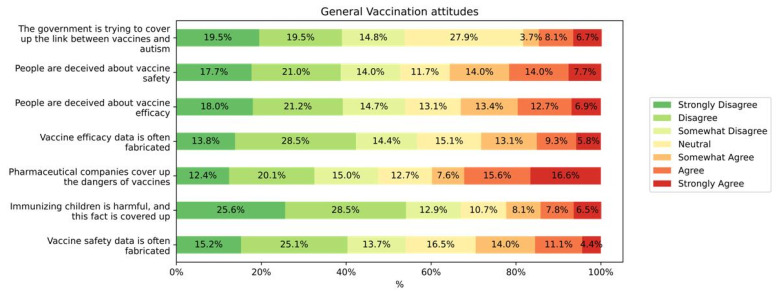
The distribution of vaccination attitudes among the participants.

**Table 1 vaccines-10-01420-t001:** The demographics of the study population.

Demographic Variable	Outcomes	Number (%)
Age	Under 20 years	106 (14.8%)
	20–29 years	534 (74.7%)
	Over 30 years	75 (10.5%)
Sex	Male	173 (24.2%)
	Female	542 (75.8%)
Marital Statue	Single	657 (91.9%)
	Married	58 (8.1%)
Residence	Urban	637 (89.1%)
	Rural	78 (10.9%)
Faculty	Natural and Life Sciences	240 (33.6%)
	Humanities	203 (28.4%)
	Economics	121 (16.9%)
	Sciences and Technology	97 (13.6%)
	Medical Sciences	54 (7.6%)
Educational level	Bachelor’s	335 (46.9%)
	Master’s	335 (46.9%)
	Post-graduate	45 (6.3%)

**Table 2 vaccines-10-01420-t002:** Logistic regression model outlining determinants predicting having heard of cervical cancer previously.

Heard of Cervical Cancer—Binary Logistic Regression (LR)						
Model Terms		$e^{\beta_{i}}$	95% CI	SE	z-Statistic	*p* Value
Intercept $(\beta_{0}$)		6.144	1.101–34.261	0.877	2.070	0.038
Sex (*p*-value: <0.0005)	Male	-	-	-	-	-
	**Female**	**2.759**	**1.728–4.406**	**0.239**	**4.253**	**<0.0005**
Age (*p*-value: 0.003)	Under 20 years	-	-	-	-	-
	20–29 years	1.585	0.868–2.892	0.307	1.499	0.134
	**Over 30 years**	**3.530**	**1.053–11.834**	**0.617**	**2.044**	**0.041**
Marital Status (*p*-value: 0.015)	Married	-	-	-	-	-
	Single	0.325	0.071–1.490	0.778	−1.446	0.148
Faculty (*p*-value: <0.0005)	Natural and Life Sciences	-	-	-	-	-
	**Economics**	**0.307**	**0.165–0.571**	**0.316**	**−3.737**	**<0.0005**
	Humanities	0.777	0.420–1.438	0.314	-0.802	0.423
	Medical Sciences	2.090	0.572–7.629	0.661	1.116	0.265
	**Sciences and Technology**	**0.431**	**0.224–0.832**	**0.335**	**−2.509**	**0.012**
Education (*p*-value: 0.001)	Bachelor’s	-	-	-	-	-
	**Master’s**	**1.853**	**1.119–3.071**	**0.258**	**2.394**	**0.017**
	Post-graduate	1.099	0.365–3.307	0.562	0.167	0.867
Log-Likelihood: −271.36	**Log-Likelihood of Null Model: −306.97**		**Log-Likelihood Ratio *p* value: <0.0005**			

Rows with significant *p* values are bolded. Please note that the *p*-value of the chi-squared testing is added below the variables.

**Table 3 vaccines-10-01420-t003:** Logistic regression model outlining determinants predicting having heard of HPV previously.

Heard of HPV—Binary Logistic Regression (LR)						
**Model Terms**		$e^{\beta_{i}}$	**95% CI**	**SE**	**z-Statistic**	* **p** * **Value**
$\mathrm{Intercept}\;(\beta_{0})$		**0.475**	**0.252–0.895**	**0.324**	**−2.302**	**0.021**
**Sex** **(*p*-value: <0.0005)**	Male	-	-	-	-	-
	**Female**	**1.700**	**1.161–2.489**	**0.195**	**2.727**	**0.006**
**Residence** **(*p*-value: <0.0005)**	Rural	-	-	-	-	-
	**Urban**	**2.038**	**1.197–3.469**	**0.271**	**2.623**	**0.009**
**Faculty** **(*p*-value: 0.006)**	Natural and Life Sciences	-	-	-	-	-
	**Economics**	**0.412**	**0.259–0.654**	**0.236**	**−3.755**	**<0.0005**
	**Humanities**	**0.482**	**0.327–0.710**	**0.198**	**−3.694**	**<0.0005**
	**Medical Sciences**	**2.452**	**1.223–4.918**	**0.355**	**2.525**	**0.012**
	**Sciences and Technology**	**0.286**	**0.168–0.484**	**0.269**	**−4.653**	**<0.0005**
**Log-Likelihood: −453.78**	**Log-Likelihood of Null Model: −493.48**		**Log-Likelihood Ratio *p* value: <0.0005**			

Rows with significant *p* values are bolded. Please note that the *p*-value of the chi-squared testing is added below the variables.

## Data Availability

The Data that supports all findings of this study are available on request from the corresponding author.

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
