# Peer review of "Knowledge and Awareness of Algerian Students about Cervical Cancer, HPV and HPV Vaccines: A Cross-Sectional Study"

_vaccines, 2022, doi:10.3390/vaccines10091420_

Round 1

Reviewer 1 Report

Thank you for the opportunity to review this cross sectional survey of post highschool Algerian students.

Inclusion criteria are not clearly stated - Line 96 students could be age 5 onward and grade could be kindergarten upwards.

No exclusion criteria were listed

Line 157- 163 Cervical screening may not actually apply to the people who answered this survey. Most countries use eligible age for screening at 25 or 30 yo and above. Authors need to put screening guidelines age for Alergia here in order to put results into context. 

Conclusions - I believe the argument is that by providing approval for HPV vaccination into the Algerian system now, this would help to decrease the risk of HPV related diseases to a current post secondary school aged population who are at risk for disease now. 

In various places in the manuscript non-scientific wording is used and  this reviewer suggests altering such language or any language that is opinion rather than fact. For example,

Line 41 "killing" - could say with half of those affected will die of disease 

Line 99 "secret" - could say that data was kept confidential or stored in a locked cabinent and locked room.

Line 330 - "HPV contaminated" is a phrase full of judgement. HPV is an exposure that is ubiguitous to anyone who has sex just like getting a cold will happen to anyone who breathes.  I would use the phrase "HPV exposed".

Line 248 "Here and interestingly" - I would delete

Line47 - Delete "HPVs"

Line  328 Knowledge - the K should be lower case

Author Response

We want to express our sincere gratitude to Reviewer #1 for the time dedicated to the review and the comprehensive, profound, and constructive remarks, which allowed us to improve the quality of our manuscript.  Below you will find responses to your comments:

  1. Inclusion criteria are not clearly stated - Line 96 students could be age 5 onward and grade could be kindergarten upwards.

No exclusion criteria were listed.

Thank you for this comment. We completely agree with these suggestions. The inclusion and the exclusion criteria have been added (lines 97-99).

  1. Line 157- 163 Cervical screening may not actually apply to the people who answered this survey. Most countries use eligible age for screening at 25 or 30 yo and above. Authors need to put screening guidelines age for Alergia here in order to put results into context. 

Thank you for this comment. According to the Algerian screening guidelines, the cervical cancer screening test is recommended and performed freely for women aged 25-65 within 3 years in public screening centers (opportunistic screening). Our survey targeted students aged 18 or over, almost 75% of whom are between 20 and 29 and 10.5% are 30 or older (table 1). Therefore, a large proportion of surveyed students are eligible for the free screening test. It is also important to note that the screening test can also be carried out in private cytology laboratories (chargeable service) where age is not specified.

  1. Conclusions - I believe the argument is that by providing approval for HPV vaccination into the Algerian system now, this would help to decrease the risk of HPV related diseases to a current post secondary school aged population who are at risk for disease now. 

Thank you for this comment. We agree with this  suggestion. The last sentence of the conclusion has been modified according to the suggestion of the reviewer (lines 354-355).

  1. In various places in the manuscript non-scientific wording is used and  this reviewer suggests altering such language or any language that is opinion rather than fact. For example,

Line 41 "killing" - could say with half of those affected will die of disease 

Line 99 "secret" - could say that data was kept confidential or stored in a locked cabinent and locked room.

Line 330 - "HPV contaminated" is a phrase full of judgement. HPV is an exposure that is ubiguitous to anyone who has sex just like getting a cold will happen to anyone who breathes.  I would use the phrase "HPV exposed".

Line 248 "Here and interestingly" - I would delete

Line47 - Delete "HPVs"

Line  328 Knowledge - the K should be lower case

Thank you for these comments. All these words were corrected according to the reviewer suggestions.

Sincerely yours

Reviewer 2 Report

The questionnaire and the participants are well matched.

Please enclose the questionnaire as an appendix.

Please illustrate the manuscript with a box containing observations/results

Please illustrate with boxes containing the conclusions drawns and perhaps, recommendations 

Author Response

We want to express our sincere gratitude to Reviewer #2 for the time dedicated to the review and the comprehensive, profound, and constructive remarks, which allowed us to improve the quality of our manuscript.  Below you will find responses to your comments:

  1. Please enclose the questionnaire as an appendix.

Thank you for this remark. The questionnaire has been added as an appendix.

  1. Please illustrate the manuscript with a box containing observations/results

Please illustrate with boxes containing the conclusions drawns and perhaps, recommendations ;

Thank you for this remark. Here we would inform you that all the authors did no well understand what the reviewer suggests exactly. Also, our manuscript was written according to the journal's guidelines and the journal do not allow to put a box in the whole manuscript or in the conclusion (if we exactly completly understand the reviewer's suggestions).

Sincerely yours